# Development and Efficacy Evaluation of a Novel Nanoparticle-Based Hemagglutination Inhibition Assay for Serological Studies of Porcine Epidemic Diarrhea Virus

**DOI:** 10.3390/vetsci12020101

**Published:** 2025-02-01

**Authors:** Fengyan Liang, Wenyue Qiao, Mengjia Zhang, Zhangtiantian Hu, Shan Zhao, Qigui Yan, Wentao Li, Yifei Lang

**Affiliations:** 1College of Veterinary Medicine, Sichuan Agricultural University, Chengdu 611130, China; fdzdyyy@163.com (F.L.);; 2National Key Laboratory of Agricultural Microbiology, Hubei Hongshan Laboratory, College of Veterinary Medicine, Huazhong Agricultural University, Wuhan 430070, China

**Keywords:** porcine epidemic diarrhea virus, serology, S1^0A^ protein, nanoparticle, hemagglutination inhibition assay

## Abstract

Porcine epidemic diarrhea virus (PEDV) is a highly contagious enteric pathogen and has caused severe economic losses to the global swine industry. In view of this, the development of rapid and accurate serological diagnostic methods is essential for the monitoring of antibody levels after PEDV infection or vaccine development. In this study, a safe and easy-to-operate method for the detection of PEDV neutralizing antibodies was constructed. The PEDV S1^0A^-Spy recombinant protein containing a sialic acid-binding motif was expressed and then presented onto SpyCatcher-mi3 nanoparticles to establish a novel virus-free hemagglutination inhibition (HI) assay. The HI assay showed good specificity and sensitivity, and the obtained HI titers correlated well with the corresponding virus neutralizing titers. It can be used as an important tool for the monitoring of PEDV neutralizing antibodies and a new way for the serological diagnosis of PEDV.

## 1. Introduction

Porcine epidemic diarrhea (PED), a swine epidemic disease caused by porcine epidemic diarrhea virus (PEDV), is characterized by severe watery diarrhea, vomiting, dehydration and high mortality in piglets [1]. PEDV is an enveloped virus with a single-stranded, positive-sense RNA genome approximately 28 kb in length. It contains seven open reading frames, which encode four structural proteins, namely the spike protein (S), envelope protein (E), membrane protein (M) and nucleocapsid protein (N) [2,3]. S is the main surface glycoprotein budded on the virus membrane of PEDV, which can be further divided into S1 and S2 subunits. The S1 subunit contains major neutralizing epitopes and mediates the binding of virus to specific host cell receptors and induces neutralizing antibodies in natural hosts, while the S2 subunit mediates the fusion of the virus and host cell membrane post virus attachment [4,5,6].

PEDV strains can be classified into two genotypes by phylogenetic analyses based on the S coding sequence: Genotype I (GI) and Genotype II (GII) [7]. The GI CV777 strain was first isolated in 1978 and epidemic in Europe [8]. In 2010, highly virulent GII PEDV strains emerged in China, causing severe PEDV outbreaks that subsequently spread across the country, resulting in a mortality rate of up to 100% in piglets and causing serious economic losses to the swine breeding industry [9,10]. Highly pathogenic PEDV was first detected in the United States in April 2013 and rapidly spread to 31 states, as well as Mexico and Canada [11]. PEDV GII quickly became the dominant strain worldwide and currently circulates in swine farms in Asia, Europe and North America, causing enormous economic losses to the swine industry and posing a potential threat to public health [12]. It has been shown that variations in the S gene, the major virulence factor, can alter the pathogenicity of PEDV, which enlarged the challenge of PEDV prevention and control in China [13,14]. In PEDV-infected pigs, the retention time of S protein antibodies in serum was longer than that of N protein antibodies due to the robust antigenicity of the S protein [15]. Furthermore, the S1 region has been shown to be a suitable target for determining genetic relationships among PEDV isolates and for developing differential diagnostic assays and effective vaccines [16,17]. Therefore, the S protein is the ideal antigen for vaccine development and the establishment of diagnostic methods.

The S1 subunit is comprised of four core domains, S1^A^ to S1^D^, whereas many alphacoronaviruses, including PEDV, contain an additional N-terminal domain, S1^0^ [18,19]. The S1^B^ domain is known to function as a receptor-binding domain for a variety of coronaviruses, such as transmissible gastroenteritis virus (TGEV) [20], middle east respiratory syndrome coronavirus (MERS-CoV) [21,22] and severe acute respiratory syndrome coronavirus 2 (SARS-CoV-2) [23]. In addition, certain PEDV strains have sialic acid (SA)-binding activity and hemagglutination activity, which were mapped to the amino-terminal 246 residues, constituted mostly by S1^0^ of the PEDV S1 subunit [6]. Li et al. identified six neutralizing or non-neutralizing epitopes in the S1 subunit of the PEDV spike protein, which further confirmed that the neutralizing epitopes may be mainly located in the region with SA-binding activity (amino acids 19-246, mostly S1^0^) or the deduced proteinaceous receptor-binding domain (S1^B^) [24].

Taking into consideration the huge threat that PEDV poses to the swine breeding industry, it is of utmost importance to establish rapid and accurate PEDV detection methods to effectively block PEDV transmission and prevent major outbreaks. Diagnostic methods for PEDV are currently divided into two categories: etiological and serological diagnostic methods. Etiological diagnostic methods are mainly based on the detection of viral nucleic acids or proteins, such as by polymerase chain reaction (PCR). Serological diagnostic methods investigate for antibodies produced by the host immune system post virus infection, such as by the indirect immunofluorescence test (IFA), virus neutralization test (VN) and enzyme-linked immunosorbent assay (ELISA) [25,26]. Serological assays determine previous pathogen exposure by identifying the presence of antibodies. Ferrara et al. used an enzyme-linked immunosorbent assay (ELISA) method to explore the transmission of porcine coronaviruses (including PEDV) and porcine reproductive and respiratory syndrome virus in southern Italy, and the results showed that PEDV was the most common coronavirus circulating in pigs in the region [27]. In the first seroepidemiological study of PEDV in Mexico, it was reported that 61.66% of the serum samples were positive, showing that PEDV is still circulating in the main pig-producing states in Mexico [28]. Serology studies of PEDV can effectively reveal the epidemic status of PEDV in certain areas, which is of great significance for PED prevention and control. The outcome of serological methods can monitor the immune status of the population, which is important for determining the infection status of a specific herd or evaluating the protecting effects post vaccination. One major hurdle for PEDV serological methods is the assessment of neutralizing antibodies elicited by PEDV infection. The VN assay is generally recognized as the “golden standard”, but its application requires culturing of contagious PEDV contained within biosafety laboratories. Therefore, the development of safe, easily operatable methods with results congruent to VN is genuinely necessitated for PEDV serology.

As a classic technique in the diagnosis of infectious pathogens, the hemagglutination assay (HA) is operatable and economically efficient and has been widely used in the qualitative or quantitative analysis of viruses with agglutinating ability, such as influenza virus [29]. The derived hemagglutination inhibition (HI) assay was henceforth developed to detect antibodies that block the virus–erythrocyte interaction [30]. Due to its high sensitivity and specificity, the HI assay has now become one of the classic tools of global influenza surveillance, and VN and HI assays are generally recognized as the “gold standards” for detecting influenza virus-neutralizing antibodies [31]. Of note, to avoid culturing of virus particles, designations of de novo HA and HI assays through presenting recombinant viral proteins on platforms such as magnetic beads or nanoparticles have been applied in numerous studies [32,33].

In the present study, we developed a novel HI assay for the detection of PEDV antibody levels through a multimeric exhibition of the chimeric PEDV S protein on self-assembling mi3 nanoparticles. Our data show that the results obtained with the newly developed HI assay are highly compatible with the VN results, which underscores the potential of large-scale application of HI assay in monitoring PEDV-specific antibody response for seroprevalence analyses or the evaluation of vaccine compliance.

## 2. Material and Methods

### 2.1. Viruses, Cell Lines and Clinical Samples

The PEDV/SC/2020 strain was isolated and preserved by the College of Veterinary Medicine, Sichuan Agricultural University. Human embryonic kidney 293 cells (HEK-293T) and African green monkey kidney epithelial (Vero) cells, preserved by the College of Veterinary Medicine, Sichuan Agricultural University, were cultured in Dulbecco’s modified Eagle medium (DMEM; Gibco, Grand Island, NY, USA) supplemented with 10% (*w*/*v*) fetal bovine serum (FBS) (ExCell Bio, Suzhou, China), penicillin (100 IU/mL) and streptomycin (100 μg/mL). PEDV-negative sera were collected from specific-pathogen-free (SPF) pigs, while PEDV-positive sera were collected from recovered pigs post PEDV infection from farms and validated by virus neutralization tests in our laboratory. From 2020 to 2022, 253 clinical serum samples were collected across four provinces in China (Sichuan, Fujian, Zhejiang and Chongqing) from pigs of different ages, including nursery pigs, fattening pigs, milking sows, breeding boars, gilts and piglets and stored at −80 °C in our laboratory.

### 2.2. Construct Design, Protein Expression and Purification

The PEDV S1^0A^ domain (amino acids 19 to 504) encoding was cloned from viral cDNA into a pcDNA-3.1 expression plasmid. Briefly, the S1^0A^ gene was cloned in frame with a sequence encoding a CD5 N-terminal signal peptide and the Fc domain of mouse IgG2a (mFc) in the C-terminal for protein purification. In order to produce the S1^0A^ protein (S1^0A^-Spy) to be conjugated to SpyCatcher, the construct of S1^0A^ was designed as described above, with the addition of a sequence encoding a Spy tag (AHIVMVDAYKPTK) at the C-terminal of the construct, which produces a fusion protein, S1^0A^-Fc-Spy. The constructed plasmid was verified by sequencing. The S1^0A^-Spy protein was expressed in HEK-293T cells. In detail, Lipo293^TM^ (Beyotime, Shanghai, China) transfection reagent was used to transfect pcDNA3.1-S1^0A^-Fc-Spy plasmid into HEK-239T cells. At 6 to 7 days post-transfection, cell supernatants containing the soluble S1^0A^ protein were harvested, and the protein was purified by protein A-affinity chromatography (Sino Biological, China) according to the manufacturer’s instructions. The purified protein was quantified by NanoDrop spectrophotometry, checked by Western blot and stored at −80 °C until further usage.

### 2.3. Construction, Expression and Purification of SpyCatcher-mi3 Nanoparticles

The gene encoding the trimeric aldolase (mi3) derived from the hyperthermophilic bacterium *Thermotoga maritima* and SpyCatcher sequence from the CnaB2 domain of the FbaB adhesion protein from the common Streptococcus pyogenes (GenBank accession no.MH425515) was synthesized by Sangon (Biotech, Shanghai, China). The gene sequence was codon-optimized for optimized *E. coli* expression and then cloned into pET28a expression plasmid. In addition, a sequence encoding the Strep-tag at the C-terminal for protein purification was introduced into the gene sequence. The pET28a expression plasmid was transformed into *Escherichia coli* strain BL21 (Vazyme, Nanjing, China). A single colony was picked into a 10 mL starter culture of LB medium containing 100 μg/mL ampicillin and incubated for 16 h at 37 °C with shaking at 200 rpm. Then, the culture was expanded in LB liquid medium containing 100 μg/mL ampicillin at 1% inoculum. At *A_600_* 0.8, cultures were induced with 0.5 mM IPTG (isopropyl β-D-thiogalactoside) and grown for 16−20 h with shaking at 200 rpm at 22 °C. After induction, the bacteria were collected by centrifugation and sonicated on ice. Then, the lysis mixture was spun down at 14,000× *g* for 30 min at 4 °C. After collecting the supernatant, SpyCatcher-mi3 nanoparticle (NP) was purified using Strep-Tactin Sepharose beads according to the manufacturer’s instructions (IBA, Germany). Purified NPs were stored at −80 °C until further use.

### 2.4. Sodium Dodecyl Sulfate-Polyacrylamide Gel Electrophoresis (SDS-PAGE) and Western Blotting (WB)

Purified proteins as described above were mixed with 5 × SDS-PAGE loading buffer (with DTT) (Solarbio, China) and boiled at 95 °C for 10 min. The proteins were separated on SDS-PAGE gel under reduction conditions and stained with Coomassie bright blue. In addition, the S1^0A^-Spy protein was loaded on an SDS-PAGE gel and subsequently electroblotted onto a polyvinylidene (PVDF) difluoride membrane. The membrane was then blocked with 5% powdered skim milk with 0.05% Tween-20 (TBST) at room temperature for 2 h and then reacted directly with the goat anti-mouse IgG HRP-conjugated (ABclonal, Wuhan, China) secondary antibody at a 1:1000 dilution for 1 h at room temperature. Finally, the protein was visualized using Pierce ECL Western blotting substrate (Thermo Fisher, Waltham, MA, USA) according to the manufacturer’s protocol.

### 2.5. Transmission Electron Microscopy (TEM)

S1^0A^-Spy was conjugated at 25 °C for 36 h with SpyCatcher-mi3 nanoparticles (NP: S1^0A^, molar ratio 1:1.5) in reaction buffer (25 mM Tris-HCl [pH 8.5], 150 mM NaCl [pH 8.5]). Subsequently, 100 μL solution was transferred into a 1.5 mL EP tube and sent to Servicebio (Wuhan, China) for TEM observation. The sample (0.2 mg/mL) was applied to freshly glow-discharged carbon 400 mesh copper grids for 2 min and blotted with filter paper. Then, the nanoparticles were stained with 8 μL 1% phosphotungstic acid (PTA) for 1 min, the staining solution was removed by filter paper and the structure of the nanoparticles was observed by transmission electron microscope after drying at room temperature.

### 2.6. Dynamic Light Scattering (DLS)

The particle size distribution of the nanoparticles in this study was measured by DLS using a Zetasizer (Malvern Panalytical, Malvern, UK). The samples were diluted 0.1% (*w/v*) with deionized water with a pH = 7.0; a total of 1 mL of the sample was absorbed and injected into the colorimetric dish followed by triplicate analysis using this apparatus with an angle of 173°. The temperature was 25 °C and equilibration time was set to 120 s. The data were analyzed using the software provided by Malvern Instruments. Particle dH was obtained from the peak of intensity (%) vs. diameter (nm) curve (PSDi).

### 2.7. Preparation of PEDV S1^0A^ Protein Mouse Polyantiserum

The purified PEDV S1^0A^ protein was used as an immunogen, and three six-week-old female BALB/c mice (Dossy, Chengdu, China) were immunized by subcutaneous injection at a dose of 100 μg per mouse. Freund’s complete adjuvant (BBI, Shanghai, China) was used for primary immunization, and Freund’s incomplete adjuvant (BBI, Shanghai, China) was used for secondary immunization. Serum samples were collected 14 days after the secondary immunization. All the animal use protocols commissioned in this study were reviewed and approved by the Animal Ethics Committee of Sichuan Agricultural University.

### 2.8. Hemagglutination and Hemagglutination Inhibition Assays with S1^0A^-Conjugated SpyCatcher-mi3 Nanoparticles

S1^0A^ was conjugated with SpyCatcher-mi3 nanoparticles to form S1^0A^-NPs, as described above. Subsequently, S1^0A^-NPs were then 2-fold serially diluted and mixed 1:1 with human erythrocytes (0.5% in PBS). Hemagglutination was assessed after incubation on ice for 2 h, and the hemagglutinating units (HAUs) were calculated for each S1^0A^-NP.

To study the HI ability of serum samples, the serum samples were first mixed with 1% human red blood cell (Immocell, Xiamen, China) suspension at a 1:1 ratio, placed at 4 °C for adsorption overnight, centrifuged to collect the supernatant, and then the serum was inactivated at 56 °C for 30 min. The treated serum samples were then 2-fold serially diluted and mixed 1:1 with PBS containing 4 HAUs of S1^0A^-NPs. The mixtures were incubated at room temperature for 30 min and then mixed 1:1 with human erythrocytes (0.5% in PBS). Hemagglutination inhibition was recorded after 2 h of incubation on ice. The maximum dilution factor of completely inhibiting erythrocyte agglutination was used as the coagulation-inhibiting titer of the serum. Each serum sample was analyzed three times independently, and the average HI titer was used. Serum samples with an HI titer higher than 1:40 were considered to be seropositive based on the collective analysis of PEDV reference serum samples.

### 2.9. Virus Neutralization (VN) Test

Serum samples were continuously diluted twice with serum-free DMEM, then added to 96-well plates, then added with the equivalent of 200TCID_50_ PEDV virus and mixed at 37 °C for 1 h. The serum–virus mixture was inoculated onto Vero cells at 37 °C for 2 h, the supernatant was discarded, washed twice with PBS, 200 μL serum-free DMEM (containing 5 μg/mL trypsin) was added and the cytopathic effect (CPE) was observed at each pore for 48–72 h. The virus neutralization titer was defined as the reciprocal of the highest serum dilution for absolute suppression of PEDV infection. Each serum sample was analyzed three times independently, and the average VN titer was used.

### 2.10. Comparison Between HI Assay and VN with Clinical Samples

Clinical serum samples were collected and tested by HI assay and VN test. Results were compared with the VN to assess the performance of HI in terms of relative sensitivity—[(true positive/(true positive  +  false negative)] × 100%—and relative specificity—[(true negative/(true negative  +  false positive)]  × 100%—and to analyze the coincidence rate (Kappa value) between the HI assay and VN. GraphPad Prism software (version 9.0) and IBM SPSS Statistics 27 software were used for the statistical analyses. Cohen’s kappa values were determined as a measure of the overall agreement. *p*-values of less than 0.05 were considered statistically significant.

## 3. Results

### 3.1. Production of S1^0A^-Spy Recombinant Protein and Spycatcher-mi3 Nanoparticles

In order to produce the S1^0A^ protein (S1^0A^-Spy) conjugated with its matching protein SpyCatcher, the Spy Tag sequence was added to the C-terminus of the mFc region (Figure 1A). This chimeric protein was then produced in mammalian cells (HEK-293T) by transfection and purification. The recombinant protein was then verified by WB and shown to migrate to its expected size, indicating the successful expression and purification of the S1^0A^-Spy protein (Figure 1B).

To create a plug-and-display nano scaffold, the coding sequence of the trimeric aldolase (mi3) derived from the hyperthermophilic bacterium *Thermotoga maritima* was synthesized, which can self-assemble into a well-organized 60 mer dodecahedral nanoparticle. To allow multivalent presentation of Spy-tagged PEDV S1^0A^, SpyCatcher was displayed on the nanoparticle surface, and the S1^0A^ protein was confirmed to bind SpyCatcher via their Spy Tag (Figure 1C). The SpyCatcher-mi3 nanoparticles (NPs) with N-terminal strep tags were appended for affinity purification. Successful expression and purification of NPs were confirmed with SDS-PAGE analysis (Figure 1D).

### 3.2. PEDV S1^0A^ Protein Can Be Displayed Successfully on mi3 Nanoparticles

The purified recombinant protein S1^0A^-Spy and SpyCatcher-mi3 nanoparticles (molar ratio 1:1.5) were incubated at 25 °C for 36 h to achieve full coupling, thus generating S1^0A^-Spy: Spycatter-mi3 (S1^0A^-NPs). As shown in Figure 2A, SDS-PAGE analysis showed that effective conjugation was achieved between S1^0A^-Spy and SpyCatcher-mi3, with the expected molecular weight migration, which confirmed the occurrence of assembly reaction and high conjugation efficiency. The shape and size of the S1^0A^-NPs were analyzed by transmission electron microscopy (TEM) to explore whether the S1^0A^-NPs were correctly assembled. The results showed that S1^0A^-Spy and SpyCatcher-mi3 were able to assemble to the particle structure of 60 polymers as regular dodecahedrons (Figure 2B). To further verify the multivalent display of the S1^0A^-Spy protein on the surface of Spycatard-mi3 nanoparticles, the particle size distribution of Spycatard-mi3 nanoparticles and S1^0A^-NPs were determined by dynamic light scattering (DLS) analysis. Figure 2C,D shows that the mean diameter of mi3 NPs is about 47.44 nm, and that of S1^0A^-NPs is about 90.11 nm, which confirms the occurrence of the conjugation, indicating that the S1^0A^-Spy protein and SpyCatcher-mi3 nanoparticles were correctly coupled through the isopeptic bond.

### 3.3. NP-Based HI Assay Has High Specificity and Is Suitable for Clinical Sample Detection

We next tested the ability of the SpyCatcher-mi3: S1^0A^-Spy nanoparticles (S1^0A^-NPs) to agglutinate human erythrocytes by using “empty” NPs as a negative control. Strong agglutination of human erythrocytes was indeed observed for NPs displaying S1^0A^ (S1^0A^-NPs), with 2048 hemagglutination units (HAUs) at a concentration of 1µM, but not for trimeric “empty” NPs (Figure 3A). The above results indicate that S1^0A^-NPs display robust hemagglutination ability and indicate that these NPs can be used for serology screening using the HI assay.

Next, we explored the inhibitory properties of virus neutralization assay-validated porcine serum samples with HI assay by using the S1^0A^-NPs. Based on the results of the HA assay, the hemagglutination titer of S1^0A^-NPs was first determined, and a PBS solution containing 4 HAUs of S1^0A^-NPs was prepared as an antigen. The reference sera were then analyzed for HI ability. As shown in Figure 3B, PEDV negative samples (*n* = 3) showed no hemagglutination inhibition (HI titer < 40), while the HA activity of S1^0A^-NPs could be inhibited by PEDV-positive serum samples (*n* = 3, neutralization titer of 1:64) with different HI titers (HI titers of 1:80, 1:160 and 1:160, respectively), indicating that the HI based on S1^0A^-NPs is specifically due to the S1^0A^–antibody interaction. In addition, polyclonal antibodies targeting PEDV S1^0A^ prepared from mice immunized with the PEDV S1^0A^ protein were also tested through VN and HI assay, resulting in a neutralization titer of 1:5 and HI titer of 1:160. VN and HI assay showed consistency in the analysis of this serum, which also indicated that antibodies specific to the PEDV S1^0A^ protein could contribute to the neutralization process of PEDV. Apparently, the S1^0A^-NP-based HI assay exhibits high specificity for the detection of PEDV antibodies, which is a desirable characteristic for accurately identifying and differentiating PEDV infections.

### 3.4. Application of S1^0A^-NP-Based HI to Clinical Samples

To further validate the S1^0A^-NP-based HI assay in the application of an analysis of PEDV antibody prevalence in clinical samples, a total of 253 porcine clinal samples were examined with the newly developed HI assay side by side with the VN assay. As shown in Table 1 and Figure 4A, the HI assay detected 54 positive samples and 199 negative samples, while the VN detected 36 positive samples and 217 negative samples (titers detailed in Appendix A). Notably, no HI-negative sample was defined as VN-positive, indicating that the HI analysis was accurate and sensitive. The coincidence rate (*Kappa* value) between the results of the two methods analyzed by SPSS Statistics 27 software was 0.759, which indicates that the two methods have good consistency (*Kappa* ≥ 0.75). The receiver operating characteristic (ROC) analysis was further utilized to analyze the test results, and the results indicate that the HI assay and VN assay are well-correlated, with the area under the ROC curve (AUC) being 0.959 (Figure 4B). The relative sensitivity of the HI assay was 100.00%, and the relative specificity of the HI assay was 91.71%, which showed high specificity and sensitivity compared with the VN assay.

## 4. Discussion

Since 2010, a large-scale PEDV outbreak occurred in China, and highly virulent Genotype II (GII) PEDV variants emerged in various regions, resulting in mortality up to 100% in piglets [9]. Therefore, effective PEDV detection and antibody monitoring methods provide an important basis for the prevention and control of PED. Currently, PEDV detection methods include etiological and serological diagnosis, among which serological detection can be used for herd immunity status monitoring and evaluation of the protective effects post vaccination for a long period of time [26]. Normally, serology methods, including indirect immunofluorescence (IFA) assay, virus neutralization (VN) assay and traditional enzyme-linked immunosorbent assay (ELISA), need to culture infectious viruses using certain facilities. Therefore, the assays using recombinant proteins as antigens are easier to operate and standardize, which is more suitable for regular laboratory diagnosis. A Bac-to-Bac baculovirus expression system was applied previously to express a truncated PEDV S1 protein, and it established an indirect ELISA method for detecting PEDV IgA antibodies, which was used for the examination of herd immunity and the evaluation of antibody levels post vaccination [34].

The classic hemagglutination inhibition (HI) assay is the derivation of hemagglutination assay (HA) for serological analysis of erythrocyte-agglutinating viruses, such as influenza virus and Newcastle disease virus [35,36]. The essence of HA and HI is the multivalent display of viral antigens, which previously could only be achieved by using live or inactivated virus particles. Here in the present study, we designed a chimeric PEDV spike protein carrying a C-terminal Spy tag (S1^0A^-Spy), which forms isopeptide covalent bonds with the SpyCatcher protein fused to the N-terminal of the self-assembling trimeric aldolase (mi3) nanoparticle (NP). The polyhedral configuration of the generated virus-like particle, designated S1^0A^-NP, was confirmed with transmission electron microscopy (TEM) and dynamic light scattering (DLS). The HA assay showed that the S1^0A^-NPs had a decent hemagglutinating activity, with an HAU of 2048 at a concentration of 1µM. The “empty” NPs do not have an HA ability, indicating that the PEDV S1^0A^ protein is key to erythrocyte agglutination. The following HI analysis showed the agglutination could be effectively inhibited through PEDV S1^0A^–antibody interaction. Utilization of this approach allows the conduction of HA or HI assays without virus propagation, which collectively minimalizes the absolute biosafety requirements. Similar HI assays were also designed for Middle East respiratory syndrome coronavirus (MERS-CoV) and influenza virus (IAV), both of which have an HA ability [32,33]. The results described in these two studies suggest that the NP-based HI assay is highly akin to HI assays performed with virus particles and could be used to differentiate antibody responses between virus genotypes.

Notably, despite being vastly reactive to antibodies, those “virus mimics” are also antigenic as potential vaccine candidates. Recent advances have shown that the artificial virus-like particles (VLPs) based on a rational design of viral proteins can elicit rapid and effective immune responses in vivo post vaccination with VLPs or VLP-coding mRNAs [37,38,39]. Therefore, the scheme for antigen exhibition on NPs used in this study could also be considered for the future “plug in and display” design of vaccine candidates.

The S1 subunit of the S protein has the highest immunogenicity among coronaviral proteins [40]. In the case of PEDV, the S1^B^ domain, also known as the collagenase equivalent (COE) domain, was first identified as the receptor-binding domain (RBD) and the main target of (neutralizing) antibodies [41,42]. Meanwhile, researchers had identified that the N-terminal 249 amino acid residues of the S1 subunit of GII PEDV, mainly comprising the S1^0^ domain, promote virus entry through adhesion to sialic acid (SA) moieties [6]. It was then confirmed that the S1^0^ domain is also a major target of neutralizing antibodies, which function through blocking the SA attachment [24]. Therefore, the antibodies detected by HI can intuitively reflect the level of S1^0^-targeting neutralizing antibodies. Recent epidemiological studies indicate that the S1^0^ domain of PEDV strains circulating in China undergoes rapid evolution, which might be associated with changes in virulence [43]. Consequently, serological analysis of the coordinative antibody responses shall be implemented, where the HI assay could serve as a versatile tool.

In the present study, we stabilized the PEDV S1^0^ by Fc-fused expression with the adjacent S1^A^ and a C-terminal Spy tag, resulting in the chimeric protein S1^0A^-Spy. This protein elicited neutralizing antibodies when used as an immunogen, and the newly developed HI assay based on this protein could distinguish PEDV-negative and -positive serum samples. In the analysis of 253 clinical serum samples, the results obtained by the HI assay are highly consistent with VN, as shown by statistical analysis. No VN-positive sample was negative for HI, suggesting that the HI assay is specific in neutralizing antibody detection. Meanwhile, one-third of the HI-positive samples (18 out of 54) is negative in VN analysis. Considering the overall results, it is unlikely that those samples contain only PEDV S1^0^ antibodies, but rather the VN assay is less sensitive than the HI, and the neutralizing antibody titers in those samples are below the detection limit. It is possible that with the rapid evolution of PEDV S1^0^, the dominant epitopes that elicit neutralizing immune responses had gradually shift from S1^B^ to S1^0^, while future epitope mapping analysis will be implemented with newly emerging PEDV strains. Taken together, due to the high specificity and sensitivity, the newly developed HI assay could be used as an alternative or complementary serology method for VN, especially when a large number of samples need to be screened within a short period of time.

## 5. Conclusions

We successfully developed and validated an HI assay for detecting neutralizing antibody levels in porcine serum samples. The results indicated that the nanoparticle-based, virus-free HI assay had good specificity and sensitivity, and the obtained HI titers have good relevancy with the corresponding VN titers. With the advantages in efficiency and biosafety, the newly established HI assay will find future utility in PEDV serology, such as in the evaluation of herd immunity and monitoring of vaccine compliance.

## Figures and Tables

**Figure 1 vetsci-12-00101-f001:**
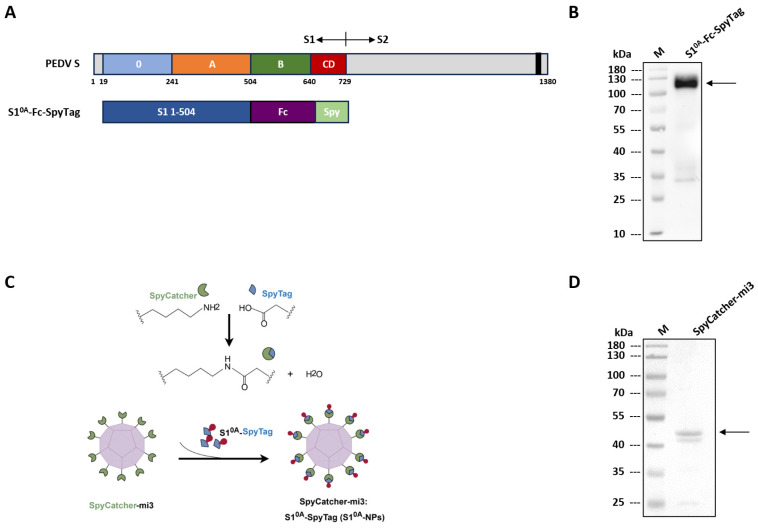
Construction and characterization of the recombinant S1^0A^-Spy protein and SpyCatcher-mi3 nanoparticles. (**A**): Schematic representation of the PEDV spike protein sequence (drawn to scale). S1 and S2 subunits of the 1380-aa-long PEDV spike protein are indicated, as well as the five domains of S1 and their respective boundaries: 0 (blue), A (orange), B (green) and CD (red). The S2 subunit is marked in dark gray. The position of the transmembrane domain (black bar) is indicated. Diagram of the S1 variant used in this study. (**B**): Affinity-purified Fc-tagged S1^0A^-Spy was identified by Western blot. M, protein marker, with numbers on the left side indicating molecular masses (in kilodaltons). (**C**): Cartoon presentation of the Plug-and-Display decoration of S1^0A^-nanoparticles (S1^0A^-NPs) by forming spontaneous isopeptide bonds between S1^0A^-Spy protein and SpyCatcher-mi3 nanoparticles. (**D**): Purified NPs were analyzed by SDS-PAGE. M, protein marker; numbers on the left are molecular masses (in kilodaltons).

**Figure 2 vetsci-12-00101-f002:**
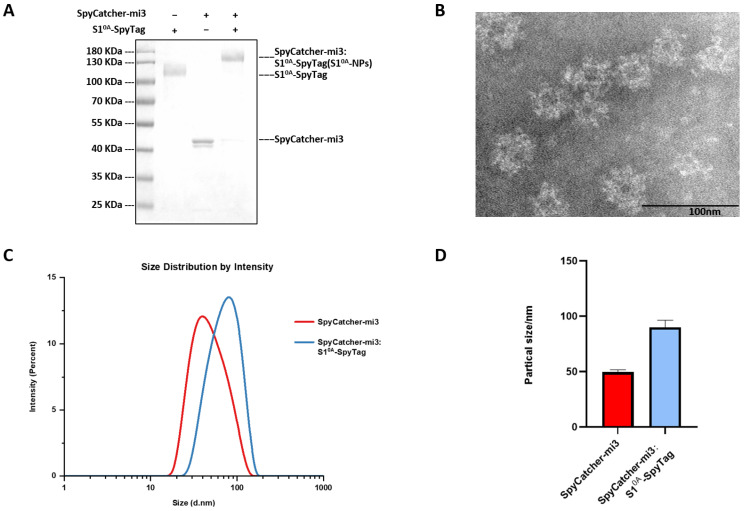
S1^0A^-Spy can be efficiently conjugated to SpyCatcher-mi3 NPs. (**A**): SpyCatcher-mi3 was incubated with S1^0A^-Spy at a 1.5:1 molar ratio at 25 °C for 36 h, followed by analysis of S1^0A^-NP formation via reducing SDS-PAGE. (**B**): TEM image of S1^0A^-NPs was negatively stained. Scale bar: 100 nm. (**C**,**D**): Particle sizes of SpyCatcher-mi3 NPs and S1^0A^-NPs were analyzed by DLS.

**Figure 3 vetsci-12-00101-f003:**
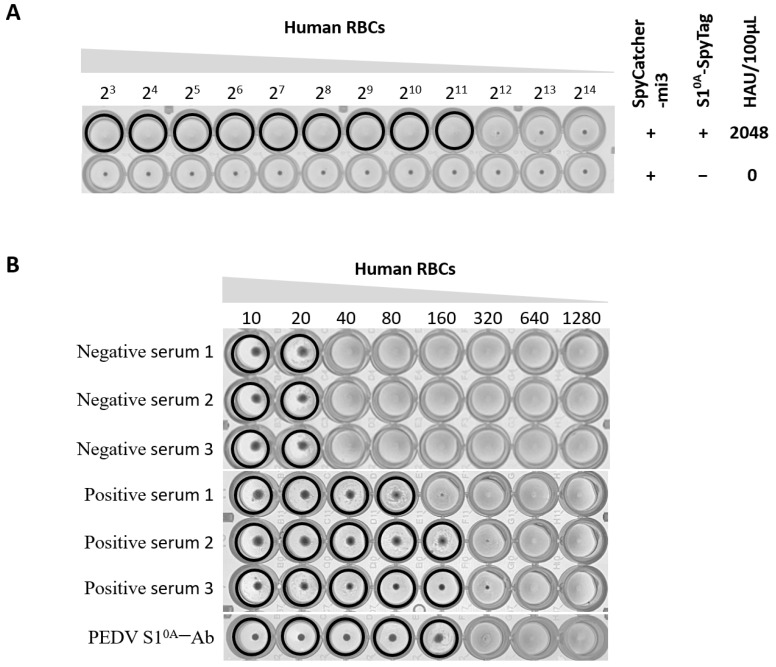
NP-based HI assay has high specificity. (**A**) Hemagglutination of human erythrocytes using S1^0A^-NPs. S1^0A^-NPs or empty self-assembling mi3 nanoparticles were serially diluted and tested for the ability to agglutinate human RBCs. (**B**) Specificity of the HI assay for the detection of PEDV reference serum. Serum samples were serially diluted and tested for the ability to block S1^0A^-NP-induced hemagglutination of human RBCs.

**Figure 4 vetsci-12-00101-f004:**
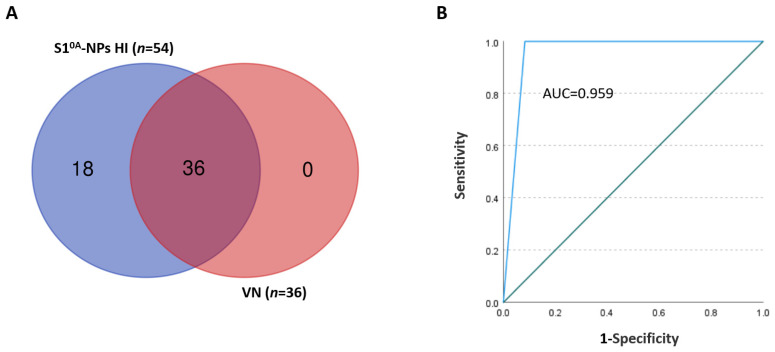
Venn diagram and ROC curve analysis of HI and VN detection results of pig serum samples. (**A**): Using Venn diagrams, the overlap between the number of positive samples in serum samples tested for HI and VN was shown. (**B**): The correlation of HI and VN to serum detection results was analyzed by the ROC curve.

**Table 1 vetsci-12-00101-t001:** Detection of anti-PEDV antibodies in pig serum samples.

Method		VN		Total
		Positive Sample No.	Negative Sample No.	
**HI assay**	**Positive sample no.**	36	18	54
	**Negative sample no.**	0	199	199
Total *Kappa* value		36	217	253 0.759

## Data Availability

All data analyzed during this study are available from the corresponding author upon reasonable request.

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
