# Peer review of "Development and Efficacy Evaluation of a Novel Nanoparticle-Based Hemagglutination Inhibition Assay for Serological Studies of Porcine Epidemic Diarrhea Virus"

_vetsci, 2025, doi:10.3390/vetsci12020101_

Round 1

Reviewer 1 Report

Comments and Suggestions for Authors

The present work describes a new nanoparticle-based approach for serological testing to detect anti-PEDV antibodies. The work is endowed with discreet methodological rigor, innovation and quality. For this reason only minor revisions are necessary for its acceptance. Below are some of my specific comments:

1) Abstract: Change the word "notorious" to a synonym and delete "breeding" from "swine breeding industry". Furthermore, the abstract should contain more detailed information about the results and how the results were obtained. 

2) Introduction: The introduction lacks some aspects, i.e. it does not specify in particular the role of serology (surveillance, evaluation of diffusion and immunogenicity) and molecular biology in the fight against PEDV and does not specify how widespread the virus is in Asia, Europe and US (countries where pork is widely raised and marketed). I recommend the authors to present the seroprevalences of the latest work carried out in pigs and wild boars: A Serological Investigation of Porcine Reproductive and Respiratory Syndrome and Three Coronaviruses in the Campania Region, Southern Italy; Seroepidemiology Study of Porcine Epidemic Diarrhea Virus in Mexico by Indirect Enzyme-Linked Immunosorbent Assay Based on a Recombinant Fragment of N-Terminus Domain Spike Protein ; nRetrospective Serosurvey of Three Porcine Coronaviruses among the Wild Boar (Sus scrofa) Population in the Campania Region of Italy; The emergence of porcine epidemic diarrhoea in Croatia: molecular characterization and serology; Porcine epidemic diarrhea virus (PEDV) introduction into a naive Dutch pig population in 2014; A persistent epidemic of porcine epidemic diarrhoea virus infection by serological survey of commercial pig farms in northern Vietnam.

3) Materials and methods: How was the virus isolated? Which territory? Outbreak in particular?

4) The statistical analysis subparagraph can be merged with the previous one

5) Methodological limitation: Why were sensitivity and specificity not calculated? Why weren't more extensive sampling or the use of certainly positive and negative samples considered? Specify these limits in the manuscript (fairly small number of samples).

Reviewer 2 Report

Comments and Suggestions for Authors

This is the review of the manuscript entitled “Development and Efficacy Evaluation of a Novel Nanoparticle based Hemagglutination Inhibition Assay for Serological Studies of Porcine Epidemic Diarrhea Virus” by Liang et al. Despite minor grammatical errors, this manuscript was well written, describing the development and validation of a nanoparticle based presentation of Porcine Epidemic Diarrhea Virus antigens for a Hemagglutination Inhibition Assay.

Minor concerns:

1. Section 2.8- Please explain how the 1:40 was selected as the cut-off for the HI assay.

2. Section 2.10 – The information for the clinical samples used needs to be provided. From where and when they were collected? Farm disease status, previous test results etc.

3. Section 2.10- Relative Sensitivity and Specificity was described here, but no data was presented in the result section.

4. Section 3.4- Please provide raw data for the SN and HI titers for the 54 positive sera, to show the titer correlation between the 2 assays. This can be in the form of a supplementary Table.

5. While the data showed great sensitivity for the HI assay, there was a lack of evidence to support the authors’ claim that the HI assay is more sensitive than the VN. For diagnostic assay evaluation, authors should present data for the limit of detection for each assay. I would suggest that authors perform a side-by-side comparison to determine the end-point titer of 3 control sera, or well-characterized known positive samples on both of the assays, that will provide the evidence for the assay sensitivity. The titer of the 18 VN-/HI+ samples should be reported.
